# Influence of Excitation Parameters on Finishing Characteristics in Magnetorheological Finishing for 6063 Aluminum Alloy

**DOI:** 10.3390/ma17112670

**Published:** 2024-06-01

**Authors:** Yiming Fang, Jinzhong Wu

**Affiliations:** School of Mechanical and Electrical Engineering, Shaoxing University, Shaoxing 312000, China; ymfang@usx.edu.cn

**Keywords:** magnetorheological finishing, low-frequency alternating magnetic field, 6063 aluminum alloy, excitation, surface roughness

## Abstract

The present work is aimed at studying the effects of the magnetorheological finishing process, using a low-frequency alternating magnetic field, on the finishing performance of 6063 aluminum alloy. The study investigates the influence of key excitation parameters such as current, frequency, excitation gap, and iron powder diameter on the material removal and surface roughness (Ra) of the finished workpiece by experiments. This study employs a single-factor experimental method, and the finish surface is analyzed by a Zigo non-contact white light interferometer. The magnetic field strength in the processing area increases with the increase in the excitation current and decreases with the increase in the excitation gap. When the current frequency is set to 1 Hz, the circulation and renewal of abrasives in the magnetic cluster is most sufficient, resulting in the optimal surface roughness value for the workpiece. According to the experimental results of the excitation parameters, more suitable process parameters were selected for a two-stage finishing experiment. The surface roughness of 6063 aluminum alloy was improved from 285 nm to 3.54 nm. Experimental results highlighted that the magnetorheological finishing using a low-frequency alternating magnetic field is a potential technique for obtaining nano-scale finishing of the 6063 aluminum alloy.

## 1. Introduction

With the rapid development of the electronics communication industry, electronic products such as mobile phones and handheld computers are required to be integrated, lightweight, and have electromagnetic shielding and rapid heat dissipation features. These requirements place higher demands on the materials used for mobile phone casings. One material, 6063 aluminum alloy, a non-magnetic material, is known for its lightweight, good thermal conductivity, as well as its strong compression resistance. It has increasingly been used in the manufacturing of casings for mobile phones and other electronic devices [1,2,3,4].

In recent years, studies on the finishing of 6063 aluminum alloy have been progressively reported. Mouliprasanth et al. performed electric discharge machining on 6063 aluminum alloy, as well as an analysis of different dielectrics in the electrical discharge finishing of 6063 aluminum alloy [5]. The surface roughness is primarily affected by peak current, then discharge on time has a considerable effect on Ra. Compared with mechanical machining, EDM is a high-efficiency machining method, but the surface quality of EDM is related to machining parameters and discharge forms, and the machining surface quality is unstable. Liew et al. investigated the effect of tool pin geometry on the micro-hardness and surface roughness of friction stir-processed–recycled 6063 aluminum alloy. Friction stir-processing is performed to make the fixture according to the shape of the workpiece, the processing universality is low, and the processing material is easy to produce deformation [6]. Syryamkin et al. investigated the influence of the degree of grinding of the 6063 aluminum alloy on its plasticity, extruding parameters, and properties of extruded profiles [7]. The grinding process can obtain high efficiency, but the surface of the workpiece is easy to crack and burn. Yin et al. studied the high-speed milling with internal cooling for 6063 aluminum alloy. The amount of erosion on the flank surface of the milling tool is more than that on the surface workpiece, and the cavitation is more severe than that of the workpiece [8]. Yi et al. studied isotropic etching finishing for 6063 aluminum alloy, where the Sa roughness of 53.5 nm reduced to 2.90 nm, and a maximum MRR of 15 µm/min was achieved [9]. Etching processing requires the use of templates for processing, and the production of templates will be affected by materials, production processes, and other factors, which are easy to cause errors, thereby reducing the processing accuracy.

The casings of aluminum alloy electronic products are only 1–2 mm thick, requiring surface roughness to reach a few nanometers without any deformation, which traditional mechanical finishing methods find difficult to meet [10,11]. Magnetic field-assisted finishing has been proven to be a low-damage [12,13,14,15], ultra-precision grinding and polishing method [16,17] that can effectively perform planar finishing such as on glass [18], polymers [19], stainless steel [20,21], copper [22,23], and alloys [24,25]. Kataria et al. developed a continuous flow magnetorheological fluid finishing process for the finishing of small holes of 6063 aluminum alloy, with a surface roughness from 7.3 µm to 10.5 nm [26]. This research confirms the feasibility of using magnetorheological finishing for the internal surface finishing of 6063 aluminum alloy, although studies on the planar finishing of 6063 aluminum alloy have not yet been reported.

In traditional magnetic grinding processes, issues such as the aggregation of magnetic abrasives, uneven distribution of non-magnetic abrasives, and deformation of the magnetic brush in contact with the workpiece, which is difficult to restore, are encountered [27,28]. These problems not only hinder achieving high-quality machined surfaces but also reduce the efficiency of magnetic grinding. In response, some researchers have suggested using pulsed magnetic field excitation [29] and vibration assistance [30,31] to enhance the efficiency of magnetic grinding. Although these methods can improve efficiency, achieving nanometer-level surface finishing is challenging because the use of high-frequency pulsed DC electromagnetic fields or additional vibrations can impact the workpiece, potentially damaging the surface. These methods also have issues with severe abrasive wear at the brush head of the magnetic brush, inability to transport abrasives from the root to the brush head, and low abrasive utilization rates. Zou and Wu proposed an ultra-precision magnetic grinding method using low-frequency alternating magnetic field excitation, which uses the changing magnetic forces generated under a low-frequency alternating magnetic field to cause the magnetic polishing module to disperse and contract in a cyclical manner within the finishing area, improving the dispersibility of magnetic particles and abrasives and solving the issue of deformations in magnetic clusters that are difficult to restore after contact with the workpiece [32]. However, as magnetic grinding finishing involves point-contact polishing, the polishing area of the workpiece is very small, leading to low finishing efficiency. Yin and Wu proposed a disc-type magnetorheological finishing method using low-frequency alternating magnetic field excitation to enhance finishing efficiency [33].

In order to solve the problems of low machining accuracy and surface defects existing in the traditional machining of 6063 aluminum alloy, this study employs a disc-type magnetorheological finishing method with low-frequency alternating magnetic field excitation to polish 6063 aluminum alloy. The study investigates the influence of key excitation parameters such as current, frequency, particle size of magnetic particles, and excitation gap on finishing performance.

## 2. Experimental Details

### 2.1. Principle and Setup

The finishing principle is shown in Figure 1. The experimental setup consists of a finishing system and an excitation system. The finishing system includes a finishing disc, the finishing disc spindle, and the workpiece shaft. The bottom of the finishing disc is affixed with a finishing pad, which carries a custom-made magnetorheological finishing fluid (composed of iron powder, grinding fluid, abrasive, and additives). The finishing disc can rotate at a controllable speed, which facilitates the circulation and renewal of the magnetorheological finishing fluid during the finishing process. The workpiece shaft can drive the workpiece to move up and down and rotate, with an output speed range of 0–1000 rpm. The excitation system consists of a variable frequency AC power supply (0–1 KHz), an electromagnetic coil (3500 turns), and a grooved magnetic pole (made of DT4 material). When AC is applied to the electromagnetic coil, an alternating magnetic field is generated above the magnetic pole. Under the action of the alternating magnetic field, the magnetorheological finishing fluid hardens into a dynamic magnetic finishing module. The combined action of the rotating workpiece, the moving finishing module, and the rotating finishing disc achieves the finishing of the 6063 aluminum alloy surface.

### 2.2. Experimental Method and Conditions

The 6063 aluminum alloy workpieces used in this experiment are similar in shape to curved mobile phone casings. They have a thickness of 12 mm, the bottom surface measures 50 mm × 50 mm, and the upper surface has chamfers that are 6 mm long. The samples of 6063 aluminum alloy were tested upon by using a CNC grinding machine made by Shenyang Machine Tool (Shenyang, China). The initial surface roughness is Ra 285 nm with a standard deviation of ±1 nm. This study employs a single-factor experimental method to conduct two sets of experiments; the first set examines the impact of crucial excitation parameters (current, frequency, size of magnetic particles, excitation gap) on the finishing results, and the second set selects more suitable process parameters based on the results of the first set for a two-stage finishing experiment. A Zigo non-contact white light interferometer is used to measure the surface roughness of the 6063 aluminum alloy before and after finishing, with a sampling area of 1.88 mm × 1.41 mm. Four points are measured in the processing region and then averaged. The measuring points (①, ②, ③, ④) are shown in Figure 2. The measurement error of the surface roughness is ±1 nm and the error in the material removal data is ±0.1 mg.

The conditions for the first set of experiments are shown in Table 1, with the experiment totaling 120 min and measurements taken every 20 min. The experiment uses an oil-based cutting fluid, with a workpiece speed of 600 rpm and a table speed of 30 rpm. The diamond abrasive is produced by TOMEI DIAMOND Company (Oyama, Japan), and the model is IRM series. The average particle diameter of diamond abrasives is 1 μm, with the irregular and blocky shape. First, the excitation current is set at 2, 4, 6, and 8 A to examine the impact of the current on the magnetic field strength and finishing results. Second, magnetic grinding is conducted under alternating frequencies of 1, 3, 5, and 7 Hz to assess the impact of the current frequency on the finishing results and determine the optimal current frequency. Third, grinding experiments are performed using iron powder with particle sizes of 1, 3, 6, and 9 μm to explore the impact of different iron powder particle sizes on the improvement rate of surface roughness and the material removal rate for 6063 aluminum alloy. Finally, finishing experiments are conducted with excitation gaps of 1, 1.5, 2, and 2.5 mm to study the effects of the excitation gap on the magnetic field strength and finishing results. Based on these results, the best process parameters for the magnetorheological finishing of 6063 aluminum alloy under a low-frequency alternating magnetic field are determined.

The conditions for the second set of experiments are shown in Table 2, with the experiment divided into two 40 min stages, and with measurements taken every 20 min. Throughout the experiment, 4 A of AC is applied, with an excitation gap of 1.5 mm. In the first stage, the frequency is 7 Hz with an iron powder particle size of 6 μm. In the second stage, the frequency is 1 Hz with an iron powder particle size of 1 μm.

## 3. Experimental Results and Analysis

### 3.1. Current

The magnitude of the current directly determines the strength of the magnetic field, which significantly impacts the finishing force. Figure 3 shows the effects of current on the surface roughness and material removal rate of 6063 aluminum alloy. At currents of 8, 6, 4, and 2 A, the obtained surface roughness values were Ra 7.69 nm, Ra 9.06 nm, Ra 13.85 nm, and Ra 37.28 nm, respectively, and the material removal rates were 18.87 mg, 17.02 mg, 15.98 mg, and 10.15 mg, respectively. The magnetic field strength was measured by a Tesla instrument. Under the currents of 8, 6, 4, and 2 A, the average magnetic field strength obtained was 362 kA/m, 261 kA/m, 185 kA/m, and 96 kA/m, respectively. As the current increases, the material removal rate for 6063 aluminum alloy gradually increases, and the surface roughness value decreases. This is because the stronger the current, the stronger the magnetic field strength in the finishing area, and the higher the finishing force. However, the experiment found that as the current increases, the electromagnetic coil heats up quickly, posing certain risks and hazards. Moreover, although better surface roughness values were obtained at currents greater than 6 A, the workpiece surface turned black. This is due to the excessive current raising the temperature in the finishing area too high, causing the magnetic clusters to frictionally heat the workpiece surface, turning it black, and affecting the overall finishing results.

### 3.2. Frequency

The impact of frequency on the improvement of surface roughness and the material removal rate is shown in Figure 4. During 0–60 min of finishing, the higher the current frequency from 1 to 7 Hz, the more significant the improvement in surface roughness. The lowest surface roughness value obtained was 72.93 nm at a frequency of 7 Hz, with a material removal rate of 14.37 mg. The finishing effects at 9 Hz were slightly lower than at 7 Hz. This is because as the frequency increases, the grinding pressure increases, and so does the material removal rate. However, at excessively high frequencies, due to the magnetic clusters vibrating too quickly, the impact force on the workpiece surface is too great, causing surface scratches. The rapid vibration of the magnetic clusters also leads to the insufficient mixing and renewal of magnetic particles and abrasives, resulting in lower abrasive utilization.

After 120 min of finishing, the best results were obtained at a current frequency of 1 Hz, with surface roughness values of 11.47 nm and a material removal rate of 16.23 mg. Between 100–120 min of finishing, all workpiece samples had a surface roughness around 40 nm. Using a frequency of 1 Hz showed unique advantages, because the lower frequency of vibration allowed for more thorough mixing and cycling of abrasive particles within the magnetic clusters, and at the same time, the vibration frequency of 1 Hz did not create excessive impact forces on the workpiece surface. Due to the high utilization of abrasives and the stability of the magnetic finishing tool, the best surface quality was achieved.

### 3.3. Excitation Gap

Figure 5 shows the impact of the excitation gap on the improvement in surface roughness and the material removal rate. At excitation gaps of 1 mm, 1.5 mm, 2 mm, and 2.5 mm, the obtained surface roughness values were Ra 16.38 nm, Ra 13.85 nm, Ra 22.26 nm, and Ra 31.41 nm, respectively, and the material removal rates were 12.01 mg, 15.98 mg, 14.33 mg, and 13.21 mg, respectively. The results indicate that as the excitation gap increases above 1 mm, both the material removal rate and the improvement in surface roughness decrease. In the finishing area, iron powder mixed with abrasives hardens into a magnetic cluster of a certain thickness under the action of an alternating magnetic field. Excitation gap and current are the most important factors affecting the field strength. The magnetic field strength in the processing area increases with the increase in the excitation current and decreases with the increase in the excitation gap. The smaller the excitation gap, the greater the compression of the magnetic clusters, increasing the shear force in contact with the workpiece surface, and thus increasing both the material removal rate and the improvement in surface roughness. Additionally, the magnetic field strength in the finishing area also decreases with an increasing excitation gap, reducing the hardening force of the magnetic mixture, and thus reducing the frictional forces, leading to lower material removal rates and less improvement in surface roughness.

However, in nanometer-level magnetorheological finishing, a too-small working gap causes the magnetic clusters to scratch the workpiece surface, affecting surface quality. Additionally, a too-small excitation gap can impede the vertical movement of the magnetic clusters, to some extent hindering the renewal of abrasives, and thus reducing finishing efficiency. Therefore, an excitation gap of 1 mm did not yield better finishing results.

### 3.4. Iron Powder Particle Size

Figure 6 displays the impact of iron powder particle size on surface roughness and the material removal. The experimental results show that during 0–60 min of finishing, as the particle size of the magnetic particles increases, both the improvement in surface roughness and the material removal rate gradually increase. This is because larger magnetic particles can produce longer magnetic clusters, resulting in an increase in magnetic force. However, at a particle size of 1 μm, the lowest surface roughness value achieved was 11.7 nm. When using an iron powder particle size of 9 μm, although a higher material removal rate was initially obtained, the final surface roughness was only 32.48 nm. This is because, for a smooth finishing surface, typically fine abrasives and iron powder sizes are chosen; larger iron powder particles can scratch the surface, affecting the finishing quality. Additionally, the dynamic magnetic clusters respond differently to the magnetic field; smaller iron powder sizes result in shorter response times and larger variations. Although the finishing force is less than that of larger iron powder magnetic clusters, the abrasive particles within the cluster are more thoroughly renewed and cycled.

### 3.5. Two-Stage Finishing

The key excitation parameters that affect finishing performance are current, frequency, excitation gap, and magnetic particle size. Based on preliminary experimental results, suitable finishing parameters were selected for a two-stage finishing experiment. Throughout the experiment, 4 A of AC is applied, with an excitation gap of 1.5 mm. In the first stage, the frequency is 7 Hz, with an iron powder particle size of 6 μm. In the second stage, the frequency is 1 Hz, with an iron powder particle size of 1 μm. As shown in Figure 7, after the first stage of finishing, the surface roughness of 6063 aluminum alloy improved to Ra 29.9 nm, and the material removal rate reached 15.95 mg. After the second stage of finishing, the surface roughness reached Ra 3.54 nm, and the material removal rate was 18.51 mg. Figure 8 shows the 3D morphology of 6063 aluminum alloy before and after finishing. From Figure 8a, it is evident that before finishing, the workpiece surface had deep grooves. After two stages of finishing, as shown in Figure 8b, the surface grooves were removed, presenting a smooth state, with a final surface roughness value of 3.54 nm. Figure 9 shows photographs of the 6063 aluminum alloy before and after finishing, where one can observe slight milling marks on the workpiece surface in the transverse direction before finishing. After finishing, the original milling marks were effectively removed. The results demonstrate that the magnetorheological finishing method using a low-frequency alternating magnetic field can achieve ultra-precision finishing of 6063 aluminum alloy.

## 4. Conclusions

(1)In magnetorheological finishing of 6063 aluminum alloy, as the current increases, both the improvement in surface roughness and the material removal are enhanced; however, excessively high currents raise the temperature in the finishing area, affecting the quality of the polished surface;(2)The vibration frequency of the magnetic clusters increases with the current frequency. At a current frequency of 1 Hz, the circulation and renewal of abrasives in the magnetic cluster are most sufficient, and it does not create excessive impact forces on the workpiece surface, resulting in the best finishing effect;(3)As the excitation gap increases from 1.5 mm to 2.5 mm, both the improvement in surface roughness and the material removal rate gradually decrease. An excitation gap smaller than 1 mm can interfere with the vertical movement of the magnetic clusters, to some extent hindering the renewal of abrasives and thus reducing finishing efficiency;(4)Larger iron powder particle sizes form longer magnetic clusters, which increase the material removal. At an iron powder particle size of 1 μm, the magnetic cluster response time is shortest, and the variation is greatest, achieving the best surface roughness value;(5)Using a low-frequency alternating magnetic field magnetorheological finishing method to polish 6063 aluminum alloy for 80 min reduced the initial surface roughness from 285 nm to 3.54 nm, achieving ultra-precision finishing of 6063 aluminum alloy.

## Figures and Tables

**Figure 1 materials-17-02670-f001:**
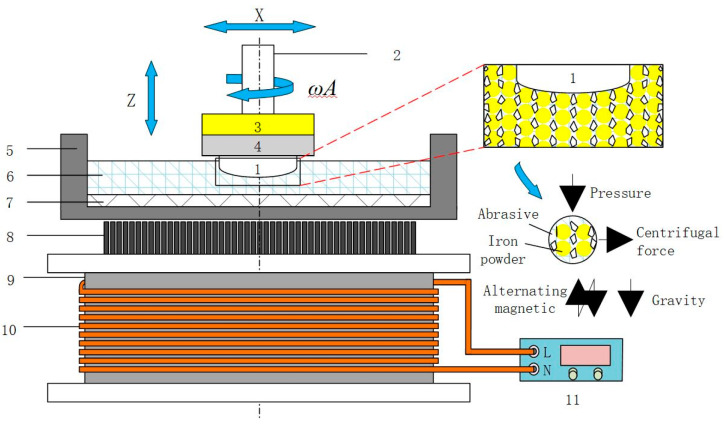
Schematic of the finishing principle: (1) workpiece, (2) axis of workpiece, (3) soft magnetic pad, (4) clamp, (5) polishing disk, (6) magnetorheological fluid, (7) polishing pad, (8) pole, (9) sleeve, (10) coil, (11) AC power supply.

**Figure 2 materials-17-02670-f002:**
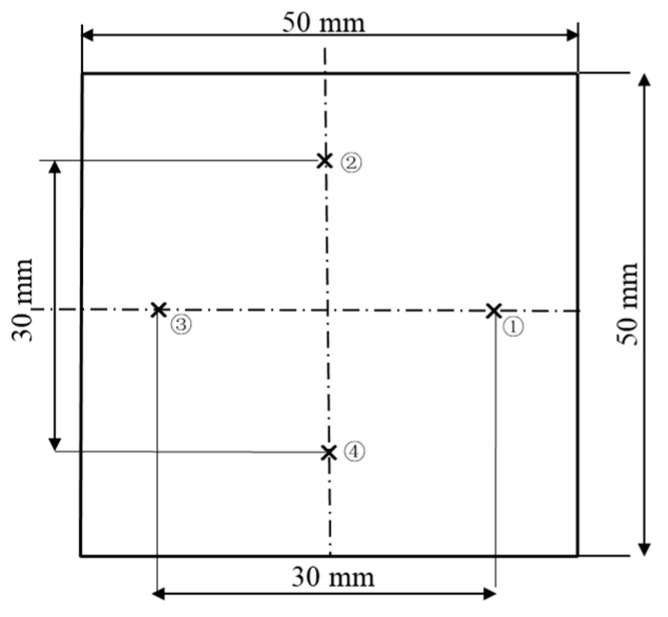
Surface roughness measuring position.

**Figure 3 materials-17-02670-f003:**
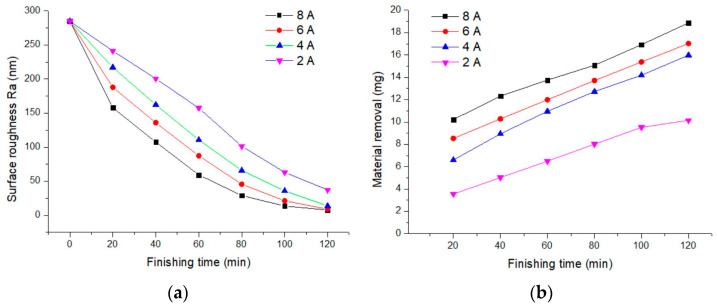
Influence of current on finishing effect. (**a**) Surface roughness; (**b**) material removal.

**Figure 4 materials-17-02670-f004:**
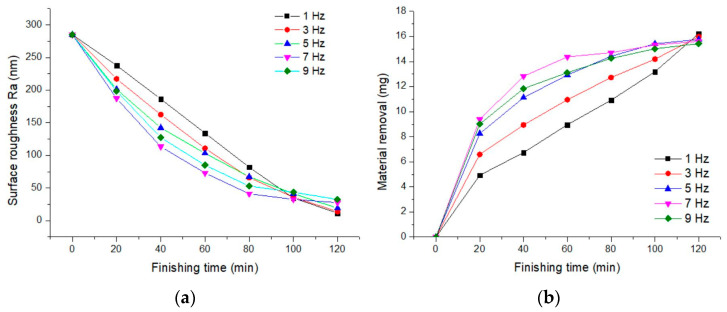
Influence of frequency on finishing effect. (**a**) Surface roughness; (**b**) material removal.

**Figure 5 materials-17-02670-f005:**
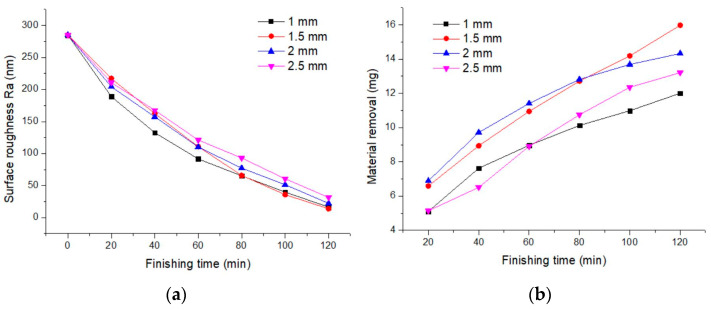
Influence of excitation gap on finishing effect. (**a**) Surface roughness; (**b**) material removal.

**Figure 6 materials-17-02670-f006:**
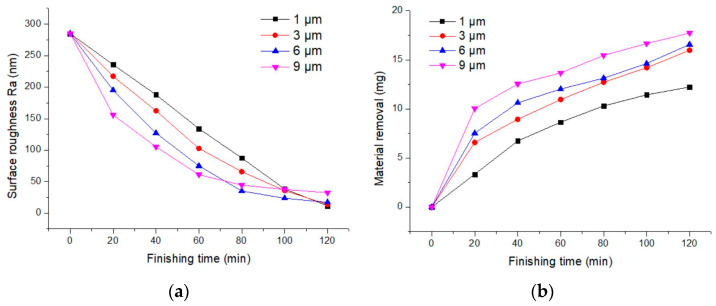
Influence of iron powder diameter on finishing effect. (**a**) Surface roughness; (**b**) material removal.

**Figure 7 materials-17-02670-f007:**
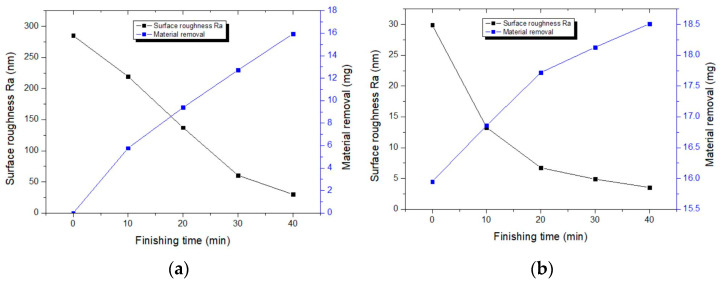
Two-stage finishing. (**a**) First stage; (**b**) second stage.

**Figure 8 materials-17-02670-f008:**
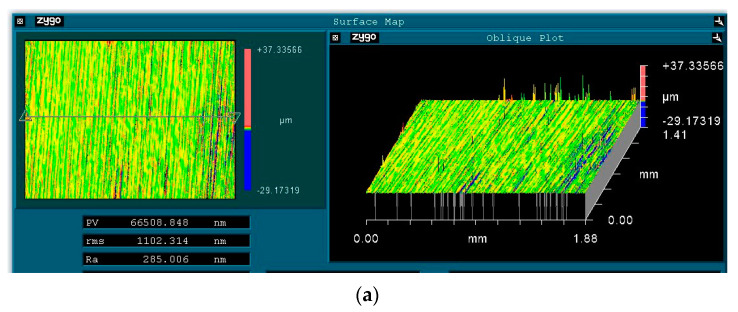
3D surfaces before and after finishing. (**a**) Before finishing; (**b**) after first stage; (**c**) after second stage.

**Figure 9 materials-17-02670-f009:**
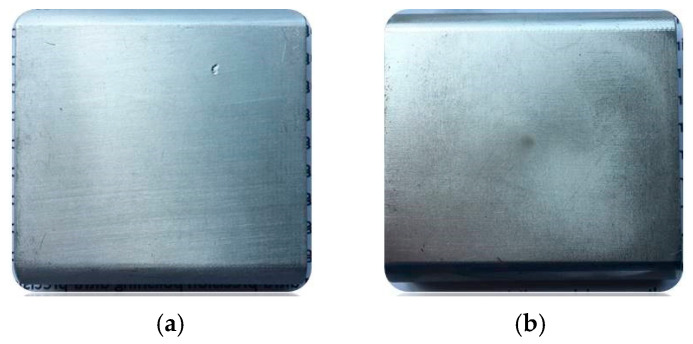
Surface contrast before and after finishing. (**a**) Before finishing; (**b**) after finishing.

**Table 1 materials-17-02670-t001:** First experimental conditions.

Parameter	1st Experiment			
	Test 1	Test 2	Test 3	Test 4
Finishing time (min)	120			
Grinding fluid (mL)	Oil-based cutting fluid			
Abrasive	Diamond, (mean dia): 1 μm			
Tray rotation speed (rpm)	30			
Workpiece rotation speed (rpm)	600			
Average alternating current (A)	2/4/6/8	4	4	4
Carbonyl iron powder (μm)	3	1/3/6/9	3	3
Excitation gap (mm)	1.5	1.5	1/1.5/2/2.5	1.5
Frequency (Hz)	3	3	3	1/3/5/7

**Table 2 materials-17-02670-t002:** Second experimental conditions.

Parameters	First Stage	Second Stage
Finishing time (min)	40	40
Average alternating current (A)	4	4
Carbonyl iron powder (μm)	6	1
Frequency (Hz)	7	1
Excitation gap (mm)	1.5	

## Data Availability

Data are contained within the article.

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
