# Peer review of "Influence of Excitation Parameters on Finishing Characteristics in Magnetorheological Finishing for 6063 Aluminum Alloy"

_materials, 2024, doi:10.3390/ma17112670_

Round 1
Reviewer 1 Report
Comments and Suggestions for Authors
In their study, the authors demonstrated an advantageous solution in the magnetorheological polishing process consisting in the use of a low-frequency alternating magnetic field, when they further related this to other basic parameters of this process, such as the intensity of the generated magnetic field (presented by means of an electric current), the frequency of the field or the size of the magnetic particles. The study appears to be of interest to the scientific community dealing with the magnetorheological polishing process, and thus, after the following comments are answered, this study may be accepted for publication in the journal Materials.
1. The Introduction section is correctly aimed at the area studied in the submitted manuscript, but in order to present the uniqueness of magnetorheological finishing, it might also be interesting for the reader to mention that the mentioned technology can be also used for the surface treatment of glass (DOI 10.1080/10426914.2021.2006222) or polymers ( DOI 10.3390/ijms232012187).
2. The manuscript should be checked from the point of view of English ("Yin et al. the high-speed", "current size" etc.) and formal pages (not defined abbreviation "EDMs etc.).
3. How many pieces of 6063 aluminum alloy workpieces were prepared by which technology (this should already be stated in the Experimental method section) and what was the standard deviation of the initial surface roughness (285 nm)?
4. In the Conditions section, the experiment duration is 100 minutes, although the results indicate a duration of 120 minutes. The diamond abrasive particles used should also be characterized in more details.
5. For comparison with other studies, it would be appropriate to present the applied magnetic field during the magnetorheological polishing process in appropriate units (e.g. kA/m), not only in A supplied to the home-made electromagnet.
6. On what basis did the authors choose a time of 40 minutes for the second set of experiments?
7. It would be appropriate to briefly discuss whether the detected dependence of process quality at different currents would be similar for other gaps as well.
8. In general, it would be appropriate to state in the study how many independent measurements were used for individual cases and whether, for example, the obtained average values are presented.
9. As part of the presentation of the results of two-stage finishing, it would be good to mention the key parameters for easier orientation of the reader.
Reviewer 2 Report
Comments and Suggestions for Authors
In this paper the authors investigated the effect of magnetorheological finishing using low frequency alternating magnetic field on the finishing characteristics of 6063 aluminium alloy samples. The investigations were focused on the impact of excitation parameters (current frequency, current size, particle size of magnetic particles, excitation gap) on mean surface roughness of specimens and material removal rate.
The paper is presented in a well-structured manner.
The graphic quality of the Figures is acceptable.
The conclusions are consistent with the arguments presented. However, the qualitative conclusions are rather well known.
The results did not provide an advancement of the current knowledge. I consider the novelty and impact of this article on the development of magnetorheological processing to be negligible. The discussion of the results is very cursory. The authors describes what was done without a deep physical analysis of why such results were obtained.
The cited references are consistent with topic studied. Relevance of cited sources is acceptable.
90% of Abstract section is devoted to results. However, main methods and treatments applied were not specified. This section requires a complete rewrite. The abstract should be a total of about 200 words maximum. The abstract should be an objective representation of the article: it must not contain results which are not presented and substantiated in the main text and should not exaggerate the main conclusions.
The Introduction section: Most of the cited works specify only what was studied in them. The main conclusions from the analysis of the cited works were not presented.
Temperature state of test material was not described.
The Introduction section is very brief, cannot present the studied problem adequately, as well as the novelty of this approach. The Introduction section should be rewritten, reflecting in it the relevance of the problem solved and the scientific novelty of the research performed based on the previously published papers.
The authors use many mental shortcuts and undefined parameters, for example: 'The initial surface roughness is 285 nm.' 'Surface roughness’ is a dimensionless property of a surface. This surface roughness is characterized by surface roughness parameters. And only those parameters defined in the relevant international standard should be used.
Characterizing the surface using only one roughness parameter is insufficient. When characterizing machine surfaces, several specific parameters are usually used, depending on the type of machining process and the nature of the resulting surface. The reviewer encourages authors to study the literature.
page 4. What is 'mean dia'?
page 5: 'Ra' IS NOT SURFACE ROUGHNESS. Check any international standard to find the name of the surface roughness parameter Ra.
On page 5, the authors point to conclusions related to the material removal rate. However, a method for determining this parameter is not provided.
Repeatability of measurements. By performing just one test you cannot draw safe conclusions, as tribological measurements are known for their variability. At least one example showing a duplicate test should be included to indicate spread of experimental values.
The study does not present a statistical analysis of the results. In tribological analysis, the graphs without error bars do not represent a reliable value.
The paper lacks discussion and analysis of results. Compare the obtained results with the results of other researchers and previous works.
According to the author contributions, Yiming Fang is the originator of the research, Jinzhong Wu developed methodology and cured data. However, it is not known who conducted the experimental research. If data has been taken from the literature, it should be clearly stated along with the source.
The English of this manuscript should be improved.
Round 2
Reviewer 1 Report
Comments and Suggestions for Authors
The authors followed reviewers’ comments appropriately and all the questions were answered or clarified. Hence, it is my pleasure to recommend current form of the paper for its publication in Materials.
Reviewer 2 Report
Comments and Suggestions for Authors
The authors introduced changes to the manuscript as suggested by the reviewers. They have now provided satisfactory answers to the reviewer's questions and comments. I would like to thank the authors for responding to my comments. The manner of revising their manuscript is satisfying, based on which the reviewer recommend the publication of this manuscript in Materials.